# The Main Arboviruses and Virus Detection Methods in Vectors: Current Approaches and Future Perspectives

**DOI:** 10.3390/pathogens14050416

**Published:** 2025-04-25

**Authors:** Amanda Montezano Cintra, Nathália Mayumi Noda-Nicolau, Milena Leite de Oliveira Soman, Pedro Henrique de Andrade Affonso, Guilherme Targino Valente, Rejane Maria Tommasini Grotto

**Affiliations:** 1Multiuser Central Laboratory, School of Agricultural Sciences, São Paulo State University (UNESP), Botucatu 18618-687, Brazil; amanda.cintra@unesp.br (A.M.C.); nathaliamnicolau@gmail.com (N.M.N.-N.); milena.leite@unesp.br (M.L.d.O.S.); pedro.a.affonso@unesp.br (P.H.d.A.A.); 2Clinical Hospital of School Medicine of São Paulo State University, São Paulo State University (UNESP), Botucatu 18618-970, Brazil; valentegt@gmail.com

**Keywords:** arboviruses, vectors, virus, zoonotic spillover, viral detection, emerging infectious diseases

## Abstract

Arthropod-borne viruses (arboviruses) represent a growing concern for global public and veterinary health, with cases reported across all continents. This review presents a broad overview of the geographic distribution of arboviruses transmitted by insect vectors, emphasizing the importance of early viral detection as a cornerstone of surveillance and outbreak preparedness. Special attention is given to the phenomenon of zoonotic spillover, where viruses maintained in natural transmission cycles often involving wildlife reservoirs and arthropod vectors cross into human populations, triggering emergent or re-emergent outbreaks. This article discusses key arboviral families of medical and veterinary significance, including Togaviridae, Flaviviridae, Nairoviridae, Phenuiviridae, Peribunyaviridae, and Orthomyxoviridae, highlighting their molecular and structural characteristics. These features are essential for guiding the development and implementation of specific and sensitive detection strategies. In addition, this work provides a comparative analysis of diverse laboratory methodologies for viral detection in vectors. From serological assays and viral isolation to advanced molecular tools and next-generation sequencing, we explore their principles, practical applications, and context-dependent advantages and limitations. By compiling this information, we aim to support researchers and public health professionals in selecting the most appropriate tools for vector surveillance, ultimately contributing to improved response strategies in the face of arboviral threats.

## 1. Introduction

Arthropod-borne pathogens contribute to more than 17% of infectious diseases, affecting millions of people around the world each year and playing a pivotal role in the emergence of new human pathogens. Dengue, the most prevalent arboviral disease, results in approximately 90 million cases and about 40,000 deaths a year [1]. Emerging arboviruses such as *Phlebovirus riftense* (Rift Valley fever virus), *Alphavirus mayaro* (Mayaro fever virus), *Orthoflavivirus nilense* (West Nile fever virus), *Alphavirus chikungunya* (Chikungunya fever virus), and *Orthoflavivirus encephalitidis* (tick-borne encephalitis virus) have also garnered scientific attention as public health concerns [2].

Arboviruses are a diverse group of more than 500 viruses transmitted by arthropod vectors such as mosquitoes and ticks [3] present on several continents (Figure 1). In the Southeast Asian region, the presence of *Orthoflavivirus denguei*, or dengue fever virus (DENV), and *Orthoflavivirus japonicum*, or Japanese encephalitis virus (JEV), probably due to anthropogenic actions and geographic changes, is the result of a “spillover” of natural zoonotic pathogens into the human population [2,4].

The South American region experiences a high incidence of infections caused by arboviruses, Mayaro fever virus (MAYV) and *Orthoflavivirus flavi*, or yellow fever virus (YFV), are predominantly found in specific biomes and population densities of rural areas; however, due to the constant urbanization of these places, there is a high risk of an increase in cases and epidemics [5,6]. DENV incidence in South American countries shows periodic oscillations [7]. According to the Pan American Health Organization (PAHO) report on the epidemiological situation of dengue in the Americas alerts, DENV cases in the year 2024 increased by 283% compared to the average of the last 5 years. With Brazil leading the highest indices of infected individuals, the detection of all four different viral serotypes in the country is noteworthy [8]. Beyond the diseases caused by the dengue fever virus, the Chikungunya fever virus (CHIKV) is also prevalent in South America, especially in Brazil, thus posing a challenge up to the present time [9]. The *Orthoflavivirus zikaense*, or Zika virus (ZIKV), has been very present in South America since 2015, and currently, with the lack of specificity of clinical presentation between cases and the complexity of laboratory diagnosis in a context of co-circulation, detecting ZIKV presents many challenges for the region’s surveillance system [6].

Rift Valley fever virus (RVFV) is endemic on the African continent and triggers large-scale outbreaks, typically associated with climatic patterns and rainfall, resulting in significant impacts on both human and animal populations [10]. The virus has spread to regions like Saudi Arabia, Madagascar, and other islands in the Indian Ocean, such as Mayotte and Comoros. Effectively managing RVFV requires addressing a range of unique challenges, involving coordination among various sectors, including public health, animal welfare, agriculture, and biosecurity [11]. Similarly, yellow fever virus (YFV) is endemic, primarily in Sub-Saharan Africa, with 274 probable cases reported by 12 countries between 2021 and 2022 [12]. The Crimean–Congo hemorrhagic fever virus (CCHFV) is also a major concern for both human and veterinary health due to its high lethality. Additionally, like many other arboviruses, environmental factors play a key role in the spread of the disease. As CCHFV is primarily transmitted by ticks, increasing human–animal interactions pose significant risks for transmission to human populations. Currently, the virus is distributed across large parts of Africa, Europe, and Asia [13]. A study on various RNA viral types highlighted the presence of unknown Thogotovirus (THOV) sequences in mosquitoes in Senegal, emphasizing the insufficient data on circulating RNA viruses. Thogotoviruses can infect a wide range of vertebrate hosts, including birds, rodents, cattle, and humans, and they remain a growing health concern in the region [14].

In the North American region, epidemiological concerns revolve around Venezuelan equine encephalitis virus (VEEV), West Nile fever virus (WNV), and La Crosse virus (LACV). VEEV first appeared in southern Texas in 1971, following an epidemic in Mexico. This infection is highly debilitating for humans and can be fatal to horses. However, the prompt actions taken by authorities, including the approval of a vaccine and mosquito control strategies, have significantly reduced the virus’s impact in the region [15]. WNV was initially identified on the African continent and has since spread to Africa, Europe, Asia, and North America. In the USA, the number of cases fluctuates annually, with nearly 44,000 confirmed and probable cases reported between 1999 and 2015 [16]. The transmission of LACV is closely linked to climatic factors and urbanization and is considered a significant pathogen with the potential to spread or emerge in new areas across North America. However, further scientific research is necessary to better understand human infections. At present, no specific treatment for LACV exists, and prevention efforts mainly focus on minimizing mosquito exposure [17].

In Europe, the circulation of WNV has been recognized since the 2000s and has been continuously monitored. Data indicate a spread of the virus on the continent and the need for vaccines to protect populations that may be at risk [18]. Tick-borne encephalitis virus (TBEV) is also a disease present on the continent, specifically in Italy, Czech Republic, and Lithuania. Studies show that despite the high incidence of the disease, there is a lack of action to combat it, either through vaccines or effective behavioral practices aimed at avoiding contact with the vector [19,20].

## 2. Arboviruses with Veterinary Importance and the “Spillover” of Natural Zoonotic Pathogens into the Human Population

The spread of viruses from other animals to humans and their risk of triggering epidemics is widely known; however, the reverse, called repercussion, is underestimated. This process can establish permanent enzootic cycles in new regions, in addition to creating a reservoir of future pandemics. Despite this, the understanding of the dynamics of enzootic arbovirus cycles is limited [21]; as multiple viruses, vectors, and hosts are involved in the transmission of arboviruses, the identification of vertebrate reservoirs is complicated [22]. The transmission of enzootic and epizootic diseases to humans occurs due to various factors, including intrinsic characteristics of the hosts, pathogens, and vectors; environmental factors such as changes in ecology and urbanization; and phylogenetic distances between the hosts, among others [1]. The accurate prediction and determination of zoonotic pathogen dissemination patterns require transdisciplinary research that integrates ecological, epidemiological, virological, immunological, demographic, and sociological data on a global scale. Only through the integrated analysis of these data using advanced computational approaches will it be possible to gain a deeper understanding of this phenomenon [23].

Among the arboviruses mentioned here, those of veterinary importance and primarily transmitted by mosquitoes include MAYV, RVFV, VEEV, WNV, JEV, and LACV, while TBEV and THOV, also present in zoonotic cycles, are transmitted mainly by ticks. MAYV was first detected in the serum of forestry workers in Trinidad [24], but it is known to infect a broad range of invertebrate and vertebrate hosts [25,26]. The virus’s transmission cycle primarily involves the *Haemagogus janthinomys* mosquito and non-human primates as primary hosts, with rodents, reptiles, and birds serving as secondary hosts [27,28]. RVFV, which has historically been present in Kenya and Tanzania, is maintained in a zoonotic cycle among African buffalo and transmitted by over 30 species of mosquitoes [29,30].

The VEE complex consists of seven species and numerous subvariants [31], with its distribution varying across the New World. Rodents act as reservoirs in a cycle involving mosquitoes of the Culex subgenus (Melanoconion), primarily in humid tropical forests and swampy areas, leading to fatal human cases [32]. WNV, maintained in a zoonotic cycle by Culex mosquitoes and passerine birds, causes outbreaks in both horses and humans. Some viral strains are found worldwide, while strain 2 remains enzootic in Africa [33]. Public health measures will continue to focus on mosquito control, education, and, hopefully, new therapeutic developments [34].

The JEV serocomplex includes WNV, Murray Valley encephalitis virus, and St. Louis encephalitis virus (SLEV). JEV was first documented in Japan in the 1800s, with its cycle maintained by Culex mosquitoes, pigs, horses, and humans [35]. The LACV virus is transmitted by *Aedes triseriatus* mosquitoes that breed in tree holes and containers, with eastern squirrels acting as amplifying hosts. The introduction of *Ae. albopictus* may alter the virus’s ecology and increase the likelihood of new variants emerging [36,37].

TBEV is primarily transmitted to humans through infected ticks, circulating between these ticks and various animals, including deer and small mammals, with infection rates correlating to tick activity [38]. For transmission to occur, a high level of viremia is required, which is typically found in mammals such as sheep, goats, horses, dogs, and rodents. Humans, however, do not develop sufficient viremia to transmit the virus to ticks and are considered dead-end hosts [39]. Similarly, THOV and Dhori virus (DHOV), also transmitted by ticks, cause a range of symptoms from mild febrile illness to meningoencephalitis. Animal reservoirs for both include banded mongooses, rodents, dromedaries, and livestock [40,41,42,43].

## 3. The Importance of Viral Detection in Insect Vectors for Predicting Emerging and Re-Emerging Viruses into Populations

Early detection of emerging and re-emerging arboviruses is a challenge for the medical and veterinary community worldwide, requiring a combination of factors and studies from various areas such as studies of the biology of vectors and enzootic hosts, meteorological and climatic surveillance models, and Geographic Information Systems. Due to growing urbanization and climate change, including increases in temperature and humidity, it is expected that there will be a greater spread of vectors and, consequently, a greater number of circulating arboviruses around the world, such as the growing expansion of *Ae. aegypti* mosquitoes into temperate regions [44].

A key factor for the early epidemiological surveillance of arboviruses is the use of adult mosquitoes using saliva collections with sugar bait; however, the infection rates of the vector are typically low by this route, limiting the sensitivity of the testing methods [45]. Testing engorged mosquitoes for both pathogens and antibodies is being explored as an alternative to direct vertebrate blood testing, which includes human patients and other reservoir animals [46]. With the decreasing costs of next-generation sequencing, metagenomics has become attractive for surveillance using various types of samples, allowing not only the detection of known sequences of mosquito-borne viruses, but also emerging viruses through de novo assembly [47]. In short, surveillance using potential vectors could provide vital information on circulation for emergency preparedness. Given finite surveillance resources, prioritization is key and is usually given to virus families with a history of emergencies, with the potential for transmission by urban mosquitoes, or viruses with high lethality rates. However, some resources must also be targeted broadly, as obscure viruses with no previous history of epidemics can emerge unexpectedly.

Beyond mosquitoes, ticks are increasingly recognized as important vectors in the detection of emerging arboviruses. Advances in next-generation sequencing have enabled the identification of novel tick-borne viruses associated with febrile illnesses in humans, highlighting their role in early warning systems for viral emergence.

In 2025, the Xue-Cheng virus (XCV), a novel orthonairovirus, was identified in northeastern China during surveillance of febrile patients with recent tick exposure [48]. Similarly, the Yezo virus (YEZV), initially reported in Japan in 2021, has since been detected in China, raising concerns about its broader distribution [49]. In another case, the Wetland virus (WELV) was discovered following severe illness in a patient bitten by a tick in Inner Mongolia [50]. These findings underscore the value of proactive viral surveillance in ticks, not only to better understand zoonotic transmission cycles but also to anticipate potential public health threats from emerging pathogens. Metagenomic approaches, in particular, have proven essential in these discoveries, allowing the unbiased detection of known and novel viruses directly from clinical and environmental samples, and playing a critical role in expanding our understanding of tick-borne viral diversity.

## 4. Different Virus Families Responsible from the Main Arboviruses

Arthropod-borne viruses, named arboviruses, are composed of RNA viruses with more than 500 described species, of which approximately 150 species are associated with diseases, belonging to six viral families: Togaviridae, Flaviviridae, Nairoviridae, Phenuiviridae, Peribunyaviridae, and Orthomyxoviridae. Considering that all of these viral families have different characteristics, the understanding behind structural and genomic features of these viruses is of paramount importance for accurate diagnosis. For this purpose, the main virus characteristics of each family are organized in Table 1, and, below, we describe the major features of these six viral families.

Alphavirus is the only genus in the family Togaviridae; the main viruses of this genus in terms of human pathogenicity are CHIKV and MAYV. They share several common characteristics, and their surfaces have the presence of glycoprotein spikes covering the alphavirion [51]. Their genome encodes different proteins that are important for replication-related and virus assembly-related functions; four non-structural proteins, which are nsP1, nsP2, nsP3, and nsP4; and six structural proteins, including capsid, E3, E2, 6K, TF, and E1. Their replication complex is formed in the plasma membrane of infected vertebrate cells by merging the virus membrane with the host endosomal membrane.

The subgenus Orthoflavivirus belongs to the family Flaviviridae and includes several important mosquito-borne viruses such as ZIKV, DENV, JEV, WNV, and YFV. They are all small and lipidic enveloped viruses, and basic capsid, which is composed of a single protein. The envelope surface contains two virus-encoded membrane-associated proteins. Viruses belonging to this genus contain three structural proteins: capsid, envelope protein (E) and either prM or M, for immature virions and mature virions, respectively; and seven nonstructural proteins: NS1, NS2A, NS2B, NS3, NS4A, NS4B, and NS5. Their replication occurs in the cytoplasm, more specifically in membrane vesicles derived from the endoplasmic reticulum, followed by the assembly of the virus and its export out of the cell by the vesicle transport pathway.

Orthonairovirus belongs to the family Nairoviridae; one of the most significant viruses in this genus, considering its human pathogenicity, is CCHFV. It has a spherical shape and is enveloped with glycoprotein spikes. They are responsible for encoding three different proteins: nucleoprotein, the viral glycoprotein precursor (GPC), and the large protein (L), which has RNA-directed RNA polymerase, helicase, and endonuclease domains. Its replication occurs in the cytoplasm, when the virus attaches to the host receptors by a glycoprotein dimer, and it is endocytosed into vesicles to the host cell.

The genus Phlebovirus belongs to the family Phenuiviridae; it includes one of the most significant pathogens for humans, the RVFV. They are enveloped spherical viruses, and their glycoproteins at the envelope surface are arranged in an icosahedral (T = 12) lattice symmetry. The genome is capable of encoding six proteins. A characteristic of all phleboviruses is the ambisense coding strategy of the S genome segment: the N protein is encoded in the negative-sense orientation on the S segment, while the NSs protein is encoded in the positive sense. However, both proteins are translated from separate subgenomic mRNAs that are transcribed from genomic or antigenomic RNA [52].

The Orthobunyaviruses belong to the family Peribunyaviridae and include several viruses like the LACV, Jamestown Canyon virus, and Snowshoe hare virus, which are named the California Encephalitis Virus Complex. They are enveloped, having spherical or pleomorphic shape [53]. The envelope of their surface has spikes composed of two glycoproteins called Gn and Gc. The genome of this genus is segmented in three molecules (S, M, and L) of single-stranded negative sense RNA and they are not polyadenylated. The S, M, and L segments are responsible for encoding the nucleocapsid protein (N), envelope glycoproteins (Gn and Gc), and the RNA-directed RNA polymerase and endonuclease (L), respectively; as well as the non-structural proteins (NSs and NSm). During cell invasion, Orthobunyaviruses attach to host receptors using Gn-Gc glycoprotein dimer, which leads to their endocytosis into the vesicles membrane; then, transcription and replication occur in the cytoplasm.

The genus Thogotovirus is classified within the Orthomyxoviridae family. Thogotovirus was one of the first viruses identified in this genus and was originally isolated from ticks in Kenya. They are enveloped and spherical viruses. They are responsible for encoding six to seven proteins from six distinct ORFs. During cell invasion, the virus attaches to the sialic acid receptor through the GP protein and is endocytosed by the host cell, followed by endosome acidification, which induces fusion of the virus membrane with the vesicle membrane. Subsequently, the encapsulated RNA segments migrate to the nucleus when transcription occurs.

## 5. Methodologies to Detect Arboviruses in Vector

Since arboviruses are transmitted primarily by mosquitoes and can cause a range of debilitating diseases in humans, including dengue fever, Zika virus infection, chikungunya infection, and yellow fever, the detection of these viruses in mosquito populations is critical for effective disease surveillance, control, and prevention [54]. Identification of arboviruses in mosquitoes provides an early warning system for potential disease outbreaks [55]. Timely detection allows health authorities to respond quickly with targeted interventions, preventing the escalation of infections and reducing the burden on healthcare systems [54].

Identifying the virus in mosquitoes is important for studying the intricate transmission cycles between them, providing better knowledge of the factors influencing disease spread, such as mosquito species, behavior, and environmental conditions [56]. Furthermore, the detection of arboviruses is relevant in vector control strategies, because accurate detection data guide the development and implementation of effective mosquito control strategies. These strategies include targeted application of insecticides, biological control agents, and genetic modification of mosquitoes to reduce their ability to transmit viruses [57].

Detecting arboviruses in mosquitoes contributes to the development of vaccines and treatments, enabling us to study the genetic makeup and characteristics of the detected viruses to design vaccines that target specific strains and variants [58]. Continuous monitoring of arboviruses in mosquitoes allows public health agencies to track disease trends, assess the effectiveness of control measures, and adapt strategies as needed. This surveillance is crucial for evidence-based decision making [57].

Over the years, several methodologies have been developed to identify and characterize arboviruses in mosquitoes, each contributing to a comprehensive understanding of disease transmission dynamics [59]. This article explores some of the primary methodologies used for virus detection in mosquitoes, such as serological assays, virus isolation, molecular techniques, next-generation sequencing (NGS), and combined methods.

Serological assays are essential tools for the detection of viral antigens or antibodies in mosquitoes (Figure 2A). These techniques indicate whether mosquitoes have been exposed to a particular virus, providing information on the prevalence of infections in mosquito populations. Mosquito samples can be used to detect virus-specific antibodies using various techniques. Enzyme-linked immunosorbent assays (ELISAs) and plaque reduction neutralization tests (PRNTs) are commonly used techniques. ELISA can identify virus-specific antigens in mosquito homogenates, while PRNT measures the presence of neutralizing antibodies in mosquito. Although relatively simple, these methods require prior knowledge of the virus being targeted and might lack sensitivity in certain contexts. PRNT is the “gold standard” of serological tests [60] and then could be used to identify virus-specific antibodies in different animal tissues. Gyawali et al. were able to detect Ross River Virus antibodies in small volumes of blood present in mosquito abdomens using the PRNT method, indicating the host exposure to arbovirus infection [61].

ELISA is an immunological method that detects and quantifies specific viral antigens in mosquito samples. This technique utilizes antibodies that bind to viral proteins, producing a measurable signal. ELISA is sensitive and widely used for the large-scale screening of mosquito populations to identify the presence of specific viruses. The ELISA method has been used to detect ZIKV [62] and DENV [63] in mosquitoes by targeting the nonstructural protein 1 (NS1) protein; additionally, this test was sensitive enough to detect the CHIK antigen from a small fraction of an infected mosquito homogenate [64], and it can also detect DENV and JEV antigens in mosquitoes *Aedes aegypti* and *Ae. albopictus* [65].

Another amplified ELISA procedure has been developed to detect the Western equine encephalitis virus (WEEV) in mosquitoes, with a sensitivity of detecting 10 plaque-forming units or greater of WEEV [66]. Furthermore, the use of antibodies against viral nucleoproteins and glycoproteins has been found to facilitate the detection of RVFV in mosquito tissues [67]. In another study [68], the authors developed two monoclonal antibodies that can detect and isolate a range of known and novel viruses in mosquito samples that identify proteins involved in viral RNA replication, and incorporated them into a high-throughput, economical ELISA-based screening system for the detection and discovery of viruses from mosquito populations.

In addition, an electrochemical impedimetric method using antibody-conjugated sensor electrodes has been developed to detect ZIKV, DENV, and CHIKV with high sensitivity and selectivity [69]. Furthermore, an antigen detection assay, not virus-specific antibodies, named The Rapid Analyte Measurement Platform (RAMP^®^), was designed to detect all DENV serotypes in mosquito pools. The data showed that it can detect geographically distinct strains of all four DENV serotypes in virus-spiked mosquito pools that contain at least 4.3 log10 plaque forming units/mL [70].

However, methodologies based on virus isolation involve the attempt to culture and propagate viruses present in mosquito samples, infecting susceptible cells with virus-containing mosquito samples, enabling researchers to isolate and identify viruses based on their cytopathic effects. This classical technique remains valuable for the discovery and characterization of viruses. However, viral isolation can be time-consuming, labor-intensive, and may yield low viral titers, limiting its practicality in large-scale surveillance [71].

Mosquito homogenates are inoculated into susceptible cell lines, and if the virus is viable, it will replicate within cells, leading to observable cytopathic effects. Virus isolation allows virus characterization and is essential for studying virus–host interactions (Figure 2B). Mosquitoes were used for virus isolation in several studies; for example, in Indonesia, a large surveillance of mosquitoes and mosquito-borne viruses was carried out, resulting in the isolation of insect Orthoflavivirus, Banna virus (BAV), new permutotetravirus, and alphamesoniviruses [72]. In Hunan Province, China, mosquito-borne arboviruses were isolated from various species of mosquitoes, including Akabane virus, JEV, and Tibet orbivirus [73]. In Tokyo, Japan, a comparison study of DENV isolation methods using different cell lines showed higher isolation rates using FcγR-expressing BHK cells [74]. BAV was also isolated from *Anopheles sinensis* mosquitoes in Hubei, China, revealing a new genotype [75]. In Yunnan Province, China, the Nam Dinh virus was isolated from mosquitoes using *Aedes albopictus* C6/36 cells [76]. The Vero cell plaque assay was reliable for detecting live WNV in pools containing up to 200 mosquitoes [77]. Furthermore, in another study, the authors described how to handle, process, and screen field-collected mosquitoes for infectious viruses by Vero cell culture assay. They isolated nine different viruses from mosquitoes collected in Connecticut, USA, and among these, five are known to cause human disease [78]. These studies highlight the importance of mosquito surveillance and virus isolation to understand the diversity and circulation of mosquito-borne viruses.

Molecular techniques have revolutionized the detection of viruses in mosquitoes by enabling the identification of specific viral nucleic acids. Reverse transcription polymerase chain reaction (RT-PCR) and quantitative real-time PCR (RT-qPCR) are widely used for their precision and sensitivity (Figure 2C). However, RT-qPCR is more commonly used because it is faster and allows multiple virus to be tested simultaneously and quantifies the virus load. Several arboviruses were detected in mosquitoes using RT-qPCR, such as RVFV [79,80]; MAYV [81]; Ilheus virus (ILHV) [82]; DENV, CHIKV, and ZIKV [83,84]; JEV [85]; and WNV [77,86].

The multiplex RT-qPCR assay has been described in several works. Some authors using primer and probe specifics simultaneously detected ZIKV, USUV, WNV, and CHIKV in large mosquito pools [87]. Another work describes the development of two different multiplex RT-qPCR assays, one for detecting YFV, JEV, WNV, and SLEV, and the other to detect DENV1, -2, -3, and -4 in mosquito pools [88]. In this sense, another study used a multiplex RT-qPCR assay to detect and serotype DENV in individual mosquito samples [89]. Rademan et al. developed a one-step multiplex real-time RT-PCR assay for the detection and distinction of Spondweni and ZIKV [90]. Furthermore, group-specific primers to detect Orthoflavivirus and Alphaviruses were designed to detect this genus virus in the mosquito pool [91]. In addition, Villinger et al. [92] identified specific arbovirus sequences from Orthoflavivirus, Alphavirus, Orthonairovirus, Phlebovirus, Orthobunyavirus, and Thogotovirus genera in mosquito pools using multiplex high-resolution melting analysis. Furthermore, RT-qPCR could be used as a screening before Sanger sequencing using universal primers for a specific genus such as Orthoflavivirus [93,94,95], Alphavirus [94,95], and Orthobunyavirus [95].

These assays have shown high sensitivity and specificity, allowing the simultaneous detection of multiple viral targets in mosquito samples and viral quantification [96]. However, they require well-equipped laboratories and skilled personnel, making them less suitable for resource-limited settings. Thus, the reverse transcription-loop-mediated isothermal amplification (RT-LAMP) method has been developed and validated for detecting viruses in mosquitoes and provides an alternative to PCR. This method requires simpler equipment and yields results rapidly, making it advantageous for field settings. The RT-LAMP allows rapid and sensitive detection of ZIKV [97,98,99] and BAV [100]. Another study used an LAMP-based virus-derived DNA (vDNA-LAMP) detection assay to successfully detect ZIKV in crude DNA purified from infected cultured cells and Aedes mosquitoes [101]. However, the sensitivity and specificity of a commercially available WNV real-time RT-LAMP assay were tested, and only 70 of 94 pools positive for real-time RT-PCR were also positive for this test [102]. On the other hand, in another study, the authors compared qRT-PCR and 2 RT-LAMP assays (based on different primer design approaches), for the detection of African and Asian/American lineages of ZIKV isolates, and showed that RT-LAMP detected 100% of the samples while RT-qPCR detected 88.88% of the samples [103].

Next-generation sequencing (NGS) has revolutionized virus detection in various samples, including mosquitoes. This high-throughput sequencing technique can identify known and novel viruses by sequencing millions of DNA or RNA strands simultaneously. NGS allows researchers to explore the entire virome of mosquito populations, providing valuable insights into viral diversity and potential emerging pathogens (Figure 2D). Mosquitoes were collected and subjected to metagenomic next-generation sequencing to detect viruses in several studies [104,105,106,107,108]. Three significantly novel viruses, with two being highly prevalent in *A. vexans nipponii*, were discovered using metagenomic analysis [109]. Moreover, metatranscriptomic sequencing was able to detect more than 70 known and novel viral species in mosquitoes [110]. Additionally, the metagenomic arbovirus detection approach used detected more than 88 viruses and found evidence of novel ZIKV variants circulating in the local mosquito population; hence, it can be a useful tool for identifying epidemic-causing arboviruses and collecting phylogenetic information on their source [111]. Furthermore, a PCR-based NGS protocol was developed, and the method allows the sequencing of lineage 2 WNV, without the need for cell culture isolation, even in cases of low virus titers [112].

Moreover, in molecular methodology, some authors have been using arrays system to detect virus in mosquitoes. The Lawrence Livermore Microbial Detection Array (LLMDA) can be able to detect multiple arboviruses of public health importance, including viruses belonging to the Orthoflavivirus, Alphavirus, and Orthobunyavirus genera [113]. Therefore, another study used a high-throughput chip array to detect arboviruses in mosquitoes, specifically identifying the presence of CHIKV in Iran [114]. Other authors developed a multiplexed Luminex array panel to detect medically important arboviruses in mosquitoes, including those of the Flaviviridae, Togaviridae, and Bunyaviridae families [115]. In this sense, another multiplexed Luminex array can detect and discriminate ZIKV, CHIKV, and DENV in mosquitoes [116]. These arrays can be valuable tools for the surveillance and management of arboviral diseases, allowing the detection of multiple viruses in the same mosquito pool and the identification of potential ecological associations between different viruses.

Lastly, there is a non-invasive analytical technique, named near-infrared spectroscopy (NIRS), that has been employed to identify viruses in mosquitoes. This methodology is based on the interaction between near-infrared light and the biological components present in the mosquito, allowing the detection of viral infections. Remarkably, NIRS is a rapid, reagent-free, and cost-effective tool that can be used as a non-invasive technique to detect and differentiate mosquito infections with ZIKV and DENV [117]. However, there are limited data on this methodology, and robust models must be developed and validated prior to use. As this methodology has the ability to predict infection based on specific chemical compounds or the actual pathogen, improving its predictive accuracy could be an excellent strategy in the identification of arboviruses [118].

As outlined in this review article, there are numerous methods available for detecting arboviruses, and the most suitable approach for each situation depends on several factors, including turnaround time, required equipment, advantages and limitations of each technique, and most importantly, the specific objective of the study. To facilitate understanding and provide a comprehensive overview of the main characteristics of these methodologies, we compiled a summary table (Table 2) highlighting their key features.

In summary, when rapid field detection of arboviruses is needed, RT-LAMP and RAMP are the preferred methods due to their speed, portability, and visual readouts. While NIRS offers the fastest screening capability, it tends to have lower specificity. Conversely, RT-qPCR remains the gold standard for high-sensitivity diagnostics and is extensively employed in outbreak surveillance. In cases involving the discovery of novel viruses or genomic characterization, NGS is indispensable despite its high cost and complexity. Additionally, LLMDA is a valuable tool for detecting multiple pathogens simultaneously, particularly in biodefense scenarios or during investigations of unknown outbreaks. In contexts requiring large-scale surveillance, ELISA offers a cost-effective solution for antibody detection, while electrochemical impedimetric biosensors provide portable, real-time results. Finally, PRNT and virus isolation are essential for confirming infectious viruses, though they require biosafety-level laboratories and extended processing times.

## 6. Conclusions

The detection of viruses in mosquitoes is crucial for understanding disease transmission dynamics and developing effective control measures. A combination of molecular, serological, and culture-based techniques provides comprehensive information on the virome of mosquito populations. Moreover, this study highlighted the importance of developing multiplex assays that allow the simultaneous detection of multiple viral targets in mosquito samples and viral quantification with high sensitivity and specificity, such as RT-qPCR and array systems. Advanced methodologies like NGS and metagenomics have opened new avenues in mosquito virology by enabling the identification of emerging and novel viruses. With ongoing research and technological advancements, these methodologies will continue to play a vital role in the surveillance and management of mosquito-borne diseases.

## Figures and Tables

**Figure 1 pathogens-14-00416-f001:**
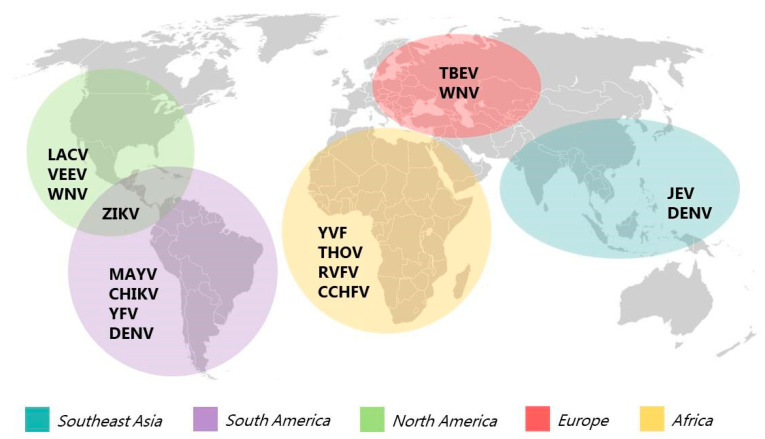
Map of arbovirus incidence. The figure highlights incident viruses and viruses related to the spillover of zoonotic pathogens from natural transmission cycles to the human population. It indicates the incidence of arboviruses in the following regions: Southeast Asia (blue), South America (purple), the African continent (yellow), North America (green), and Europe (pink). Arboviruses covered include *Orthoflavivirus denguei* (DENV), *Orthoflavivirus japonicum* (JEV), *Alphavirus mayaro* (MAYV, *Alphavirus chikungunya* (CHIKV), *Orthoflavivirus flavi* (YFV), Crimean–Congo hemorrhagic fever virus (CCHFV), *Phlebovirus riftense* (RVFV), Venezuelan equine encephalitis virus (VEEV), *Orthoflavivirus nilense* (WNV), *Orthoflavivirus encephalitidis* (TBEV), *Orthoflavivirus zikaense* (ZIKV), La Crosse virus (LACV), and Thogotovirus (THOV).

**Figure 2 pathogens-14-00416-f002:**
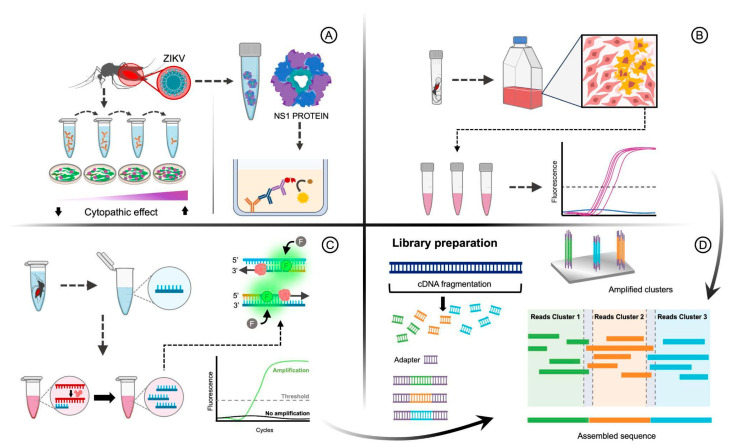
This is an illustration regarding the techniques we will address in this article: (**A**) serological assays, (**B**) virus isolation, (**C**) molecular techniques, (**D**) next-generation sequencing (NGS), and combined methods. The serological assays (**A**) are represented in two different techniques in the figure. The left side represents the plaque reduction neutralization test (PRNT) technique, which measures the presence of neutralizing antibodies in mosquito through dilution methods and inoculation into responsive cell cultures for the detection of specific viruses, observed through cytopathic effects. On the right side of the figure, the enzyme-linked immunosorbent assays (ELISAs) are depicted; they aim to identify virus-specific antigens in mosquito homogenates, involving the fixation of these proteins on a plate and detection through enzymatic reactions, resulting in a color change indicating the presence of the protein of interest (in this case, the presence or absence of the virus). The virus isolation (**B**) is represented in the figure from infected vector mosquitoes followed by homogenization of the insects, filtration of the material to remove debris, inoculation of the filtrate into cell cultures to enable viral replication, identification and confirmation of viral infection, isolation of the virus from the cultures, and molecular characterization of the virus for precise identification. Molecular techniques (**C**) are represented by RT-qPCR for the diagnosis of arboviruses in vector mosquitoes, starting with sample preparation and viral RNA extraction. Following conversion of viral RNA into complementary DNA (cDNA) through reverse transcription, the cDNA is amplified using qPCR with specific primers for the arbovirus in order to quantify the viral load in the sample. The final figure represents the NGS technique and combined methods (**D**) for arbovirus identification in vector mosquito samples, which involves several steps. Initially, vector mosquitoes are collected and species are identified. Samples are prepared, and viral nucleic acid is extracted and converted into cDNA. This material is used to create a sequencing library (a process that may include tagging viral genetic material with adapter insertion and indexing), which is then subjected to next-generation sequencing, generating sequence reads. The reads are processed by bioinformatic analysis to assemble viral sequences and identify the present arbovirus, often by comparison with reference sequences.

**Table 1 pathogens-14-00416-t001:** The main virus characteristics of Togaviridae, Flaviviridae, Nairoviridae, Phenuiviridae, Peribunyaviridae, and Orthomyxoviridae.

Viral Family	Shape	Diameter	Genome	Genome Size
Togaviridae	Spherical	70 nM	+ssRNA	9.7–11.8 kb
Flaviviridae	Spherical	50 nM	+ssRNA	9.2–11 kb
Naioroviridae	Spherical	80–120 nM	3 segments −ssRNA	Small 2 kb Medium 5 kb Large 12 kb
Phenuiviridae	Spherical	80–120 nM	3 segments −ssRNA	Small 1.7 kb Medium 3.2 kb Large 6.4 kb
Peribunyaviridae	Spherical	80–120 nM	3 segments −ssRNA	Small 1 kb Medium 4 kb Large 6.8 kb
Orthomyxoviridae	Spherical	80–120 nM	6 segments −ssRNA	10 kb

Notes: +ssRNA (positive-sense single-stranded RNA); −ssRNA negative-sense single-stranded RNA.

**Table 2 pathogens-14-00416-t002:** Arbovirus detection methods in mosquitoes: comparative features.

Method	Time	Equipment	Advantages	Disadvantages	Main Applications
ELISA	2–4 h	Microplate reader	High-throughput, detects proteins	Lower sensitivity	Antibody detection, surveillance studies
PRNT	5–7 days	Cell culture, biosafety laboratory	Gold standard for neutralizing antibodies	Labor-intensive, requires live virus	Differentiates viable viruses, serological surveys
Electrochemical Impedimetric	30 min	Portable sensor	Label-free, real-time, portable	Requires sensor optimization	Field biosensors, smart traps
RAMP	15–30 min	Water bath/heat block (37–42 °C)	Faster than LAMP, room temp possible	Less validated, limited commercial kits	Rapid field diagnostic
Virus isolation	3–14 days	Cell culture, biosafety laboratory	Confirms infectious virus, gold standard for viability	Slow, laborious, biosafety risks	Research, vaccine development
RT-qPCR	1–3 h	Real-time PCR machine	Quantitative, high sensitivity, multiplex capable	Expensive reagents	Outbreak monitoring, diagnostic
RT-LAMP	30–60 min	Heat block	Isothermal, Field-deployable, visual readout	Primer design complex, false positives	Point-of-care testing, field surveillance
NGS	1–3 days	Sequencer, bioinformatics	Comprehensive genomic data, novel pathogen	High cost, complex data analysis	Virus discovery
LLMDA	6–24 h	Microarray scanner	Detects thousands of pathogens simultaneously	Slow, expensive	Biodefense, unknown pathogens
NIRS	1–5 min	Portable NIR spectrometer	No sample preparation, rapid, reagent-free	Lower specificity, needs calibration	Mass mosquito screening

## Data Availability

All data produced during this study are included in the published article.

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
