# Peer review of "The Main Arboviruses and Virus Detection Methods in Vectors: Current Approaches and Future Perspectives"

_pathogens, 2025, doi:10.3390/pathogens14050416_

Round 1
Reviewer 1 Report
Comments and Suggestions for Authors
Comments on The Main Arboviruses and Virus Detection Methods in Vectors: Current Approaches and Future Perspectives.
This manuscript summarizes descriptions of some arboviruses causing disease in the world and the various methods of virus detection. The manuscript is well written and provides a comprehensive revision of the topic.
The Abstract could provide more information about their work that may be more appealing to readers.
This work will be more valuable if the authors provided recommendations on what to use for testing on what situations.
Given that this is work on methods, more details on how the methods work would be helpful to students.
Lines115-116 appear as disconnected from previous paragraphs.
Figure 1 legend. It would help if the authors explained why they picked some viruses and not others. Perhaps, the viruses portraited in the figure are the ones causing more cases/deaths in the world?
Line 286. Reference to Figure 3 should be Figure 2 instead?
Author Response
Comments 1: The Abstract could provide more information about their work that may be more appealing to readers.
Response 1: Thank you for pointing this out. We agree with this comment. Therefore, the abstract has been revised to more accurately reflect the scope of the manuscript, emphasizing the global distribution of arboviruses, the importance of viral detection in arthropod vectors, and the relevance of zoonotic spillover. We also included a brief overview of key viral families and highlighted the comparative analysis of laboratory detection methods, aiming to support decision-making in surveillance strategies. These changes ensure the abstract better aligns with the objectives and content of the review. The updated corresponds to page 1, lines 12 to 30.
Comments 2 and 3: This work will be more valuable if the authors provided recommendations on what to use for testing on what situations. Given that this is work on methods, more details on how the methods work would be helpful to students.
Response 2 and 3: Thank you for pointing this out. We agree with this comment. Therefore, we have added further details regarding the methods discussed in this article, page 10 (lines 322-325), page 12 (lines486-492) and page 13(lines 501-513). Additionally, we have included a table summarizing the main detection methods along with their key characteristics, which aims to assist researchers in selecting the most appropriate approach for their specific needs (Table 2, page 12).
Comments 4: Lines115-116 appear as disconnected from previous paragraphs.
Response 4: Thank you for pointing this out. We agree with this comment. Therefore, the information previously located on lines 115 and 116 has been revised to ensure better integration with the surrounding paragraphs. The updated content is now highlighted and corresponds to page 4, lines128 to 133.
Comments 5: Figure 1 legend. It would help if the authors explained why they picked some viruses and not others. Perhaps, the viruses portraited in the figure are the ones causing more cases/deaths in the world?
Response 5: Thank you for pointing this out. We agree with this comment. Therefore,the figure highlights both incident viruses and those related to the spillover of zoonotic pathogens from natural transmission cycles into human populations. This information has been added to the caption of Figure 1, located on page 3, lines 108 and 109.
Comments 6: Line 286. Reference to Figure 3 should be Figure 2 instead?
Response 6: Thank you for pointing this out. We agree with this comment. Therefore, some citations in the text referring to Figure 2 were incorrect; therefore, all figure captions and references throughout the text have been reviewed. This includes those on page 8 (lines 314 and 359), page 9 (line 379) and page 10 (line 420).
Reviewer 2 Report
Comments and Suggestions for Authors
The mansucript submitted by Amanda Montezano Cintra et al. is a litterature review that provide information on techniques used for a comprehensive understanding on the virome of mosquito populations. It provide very useful information so the authors could add information on the algorithm used for the papers included in this review and criteria for reseach of documents.
Author Response
Comments 1: The mansucript submitted by Amanda Montezano Cintra et al. is a litterature review that provide information on techniques used for a comprehensive understanding on the virome of mosquito populations. It provide very useful information so the authors could add information on the algorithm used for the papers included in this review and criteria for reseach of documents.
Response 1: Thank you for pointing this out. We agree with this comment. Therefore, while the objective of our work was not to perform a systematic review, we did conduct a structured literature search to support the development of this narrative review. Specifically, we used PubMed and Web of Science databases to identify relevant publications, using the search terms “arboviruses detection”, and “virus detection in mosquitoes”. Additionally, we included epidemiological data from official reports issued by the World Health Organization (WHO) and the Pan American Health Organization (PAHO). Our intention was to gather representative and up-to-date studies that illustrate the range of methodologies currently applied in arbovirus detection in mosquito populations.
Reviewer 3 Report
Comments and Suggestions for Authors
The review is well written and provides a comprehensive overview of different methods used to detect arboviruses in vectors. The first two chapters deal with all arboviruses, but from the third chapter onwards (The importance of viral detection in insect vectors for predicting emerging and re-emerging viruses into populations) the focus is only on mosquito-borne viruses. The authors would do well to cite some metagenomics studies on tick-borne virome as an example in addition to the studies conducted on mosquitoes. The recent discoveries of pathogenic viruses, such as wetland virus and Xue-Cheng virus, show that ticks also need to be monitored.
Author Response
Comments 1: The first two chapters deal with all arboviruses, but from the third chapter onwards (The importance of viral detection in insect vectors for predicting emerging and re-emerging viruses into populations) the focus is only on mosquito-borne viruses. The authors would do well to cite some metagenomics studies on tick-borne virome as an example in addition to the studies conducted on mosquitoes. The recent discoveries of pathogenic viruses, such as wetland virus and Xue-Cheng virus, show that ticks also need to be monitored.
Response 1: Thank you for pointing this out. We agree with this comment. Therefore, as a result of this suggestion, we added to the chapter “The importance of viral detection in insect vectors for predicting emerging and re-emerging viruses into populations” viruses such as Xue-Cheng virus (XCV), Yezo virus (YEZV), and Wetland virus (WELV), which have been detected in ticks through metagenomic studies. The modification was made and highlighted on page 5, lines 193 to 207.